# The Implications of EU Regulation 2016/429 on Neglected Diseases of Small Ruminants including Contagious Agalactia with Particular Reference to Italy

**DOI:** 10.3390/ani10050900

**Published:** 2020-05-22

**Authors:** Guido Ruggero Loria, Luigi Ruocco, Gabriele Ciaccio, Francesco Iovino, Robin A. J. Nicholas, Silvio Borrello

**Affiliations:** 1Istituto Zooprofilattico Sperimentale della Sicilia Via Gino Marinuzzi 3, 90129 Palermo, Italy; guidoruggero.loria@izssicilia.it (G.R.L.); gabriele.ciaccio@izssicilia.it (G.C.); 2Direzione Generale della Sanità Animale e dei Farmaci Veterinari-Ministero della Salute Viale Giorgio Ribotta, 5-00144 Rome, Italy; l.ruocco@sanita.it (L.R.); s.borrello@sanita.it (S.B.); 3Ministero della Salute, Ufficio Veterinario Adempimenti Comunitari UVAC, Via Cavour 106, 90133 Palermo, Italy; f.iovino@sanita.it; 4The Oaks, Nutshell Lane, Upper Hale, Farnham Surrey GU9 0HG, UK

**Keywords:** animal health law, livestock diseases, small ruminants, Italy

## Abstract

**Simple Summary:**

This article explains the future application of the EU regulation Animal Health Law (Reg. CE 429/2016), which will be implemented in 2021. The article describes the major changes and modifications related to notifiable diseases of sheep and goats and outlines the approach that will be taken for their surveillance and control. It explains the objective of the European Commission in applying a modern animal health system based on risk assessment, updated scientific knowledge, epidemiology, and a new culture of European diseases prevention. Five categories of diseases are recognized. Category A diseases are not usually present in the EU, which must be immediately eradicated. Category B diseases must be controlled in member states and eventually eradicated while category C diseases are present in some MS, which must be prevented from spreading to disease-free states. Category D diseases only present moderate risk but necessitate measures such as movement control to prevent their spread among MS or their entry into the EU, and category E diseases need surveillance. Lastly, differences between the previous Italian regulations and the new EU law are highlighted.

**Abstract:**

After almost 40 years, the 27 member states (MS) of the European Union (EU) will comply with the European Law 429/2016 in 2021 by completing a process of unification and harmonization of all regulations related to animal health between MS. These new provisions are based on modern scientific principles on animal health, on long-term epidemiological data, and, above all, on the most current risk assessment and analysis. The paper describes all changes and updates, which will impact the Italian current National regulation. A total of 58 animal diseases have been included in the Annex II (“Listing”) and Annex IV (“Categorization”) of the new Delegated Act (DA 2018/1629). Five diseases comprising the great viral epizooties were automatically included on the list because of their primary importance. These diseases include foot and mouth disease (FMD), African swine fever (ASF), classical swine fever (CSF), highly pathogenic avian influenza (HPAI), and African horse sickness (AHS). Another 53 diseases have been identified by the ad hoc assessment on listing and categorization of animal diseases developed by the European Food Safety Association. Seventeen communicable diseases of the Order Artiodactlya (sheep, goats, deer, etc.) have been listed including foot and mouth disease, sheep and goat pox, and pestes de petits ruminants. In addition, other endemic diseases affecting more than one species include blue tongue, tuberculosis, brucellosis, and anthrax. There are five categories (A-E) based on the degree of action to be undertaken throughout the EU for each disease. These vary from complete eradication for diseases not normally found in the EU like FMD (category A) for establishing surveillance for diseases like West Nile that present high risk but lack control tools (category E).

## 1. Introduction

The globalization of markets has led to a huge increase in the exchange of animals and their products for food consumption. However, this has been accompanied by serious risks of disease outbreaks such as bovine spongiform encephalopathy, avian influenza, and blue tongue (BT) [1]. In response to these threats, the European Commission (EC) has developed an up-to-date strategy of animal disease management based on a "One Health" approach founded on current scientific knowledge. This covers the link between veterinary and human health, the environment, food and feed safety, safety food supply, and economic, social, and cultural factors.

The EC’s objective was not to issue regulations for all transmissible animal diseases indiscriminately but to concentrate on regulation where subsidiarity was needed, i.e., in all those cases where a supranational approach was required to provide added value. This new European regulation aimed to produce a proper balance of competences between the European Union (EU) and the Member States (MS).

The recent Animal Health Law (AHL) EU 2016/429 [2] issued by the European Parliament and Council was passed on 31 March 2016 and set out a new, robust legal framework for animal health in Europe. The AHL brings together, in a single act, all the provisions applicable to animal trade within the EU, the entry of animals, and products into the EU, the eradication of diseases, veterinary checks, notification of diseases, and financial support. The next application of the AHL, scheduled for 2021, will modify or repeal 38 decisions, directives, and regulations adopted between 1964 to the present day on animal health.

Crucially, the AHL is linked to an epidemiological approach based on risk backed by scientific evidence that will be more dynamic. Changing and evolving as necessary, this law will provide appropriate margins of flexibility by enabling the modification of any action when new evidence emerges. The AHL allows preventive actions to be timely and more effective in the control of endemic diseases within the EU and, importantly, to exotic epizootics coming from third world countries. This will allow specific rules to be applied when confronted with new diseases. In addition to the areas already under the current legislation, specific areas of interest are regulated for the first time, such as responsibility for animal health, biosafety, epidemiological investigations, surveillance, and applications of a disease-free status. 

However, the articles contained in the AHL cover only the general rules governing animal health so it is necessary to complete the legislative process with specific acts such as the Delegates and Executive Acts, which will be essential to finalize strategies and timescales and determine the intensity of veterinary actions. The first two acts issued for the surveillance and control of animal transmissible diseases are Annex II (Diseases listing-List of diseases of European competence) [3], which came into force on 20 April 2019, and Annex IV (Diseases categorization-categorization of diseases in groups, organized by intervention priorities) [4].

## 2. Diseases Covered

Coming from the general list of animal diseases updated by the OIE (World Organisation for Animal Health) [1], a total of 58 candidate diseases were included by the MS in the Annex II [3] and Annex IV [4]. Of these diseases, 29 had already been selected for their importance during ad hoc assessment by a panel of animal health and welfare experts under the auspices of the European Food Safety Association (EFSA) [5]. Five diseases were automatically included because they are described directly in Article 5 of the AHL Regulation because of primary importance (the great viral epizooties: foot and mouth disease (FMD), classical swine fever, African swine fever, African horse sickness, and highly pathogenic avian influenza (HPAI)). Another 24 were derived from internal evaluations by the Commission and from other EFSA opinions and/or included the following notifications by the European Reference Centres (including 18 diseases of aquatic species, bees, and amphibians).

Concerning the communicable diseases of sheep and goats, 17 diseases have been included, as shown in Table 1. The serious exotic epizootics, artiodactyls of the Artiodactyls, comprise FMD, rinderpest, Rift Valley fever, and rabies. Exotic virosis comprise sheep and goat pox. Pestes de petits ruminants (PPR) within the "multi-species" category for other endemic diseases in Italy are blue tongue, tuberculosis, brucellosis, and anthrax. Other recognized diseases of community interest such as epididymitis of the ram, Q fever, and paratuberculosis have a low veterinary impact within the EU and are generally easily treatable.

## 3. Category of Diseases

The great novelty represented by this risk approach is that it covers the whole European territory. It describes the intensity of the veterinary actions to be undertaken based on the designated category of the disease (A–E), as established by the ad hoc assessment on listing and categorization of animal diseases developed by EFSA.

Each category includes a different number of animal diseases to which the same rules will apply.

Category A: This group, mostly covering diseases that do not normally occur in the EU, require the immediate and obligatory adoption of eradication measures as soon as they are identified (e.g., the greatest viral epidemics like FMD, PPR, or other exotic epizootic diseases such as caprine contagious pleuropneumonia (CCPP).

Category B: This group requires continuous monitoring in all MS so that they can be compulsorily eradicated from the EU when confirmed (e.g., diseases subject to national plans still under eradication such as tuberculosis (TB) and brucellosis).

Category C: These diseases are only present in some MS but for which measures are necessary to prevent their spread to other MS that are officially free (such as BT) or that have eradication programmes for the listed disease concerned.

Category D: These diseases only present moderate risk but necessitate measures such as movement control to prevent their spread among MS or their entry into the EU from third states (e.g., diseases of economic impact such as ovine epididymitis and echinococcosis, which are limited to particular sectors).

Category E: This group presents variable but, occasionally, high risk diseases, but for which there are no control tools because they are generally linked vectors or wildlife reservoirs such as Q fever or West Nile disease. These diseases require surveillance inside the EU.

To fall into the first three categories of A, B and C, a disease must, in addition to having to automatically satisfy the measures provided by category D (regulation of animal movements) and E (surveillance), be bound by all the epidemiological criteria identified during the EFSA assessment [5]. These comprise presence, transmissibility, transmission, susceptible species, and morbidity/mortality. The disease must also have a significant impact on at least one of the additional parameters included during the EFSA assessment, which are zoonotic risk, the economic impact, and any effects on the labour market, animal welfare, environment, and/or biodiversity.

For all the diseases categorized in A–E, surveillance activities (category E) are, therefore, always applicable with the next degree, category D. Movement control measures will be added, if necessary. The monitoring and control of movements (i.e., D and E) can be added as additional measures depending on whether the disease falls into the different categories of A, B, or C. These last three categories are mutually exclusive because each provides different criteria: a disease that falls in category A cannot be in B or C. However, the criteria of D and E always apply as transversal measures.

For sheep and goat diseases, a clear example of the new risk-based approach is the classification of BT in category C. This disease was, until recently, considered exotic. Therefore, it was potentially classifiable as A. Instead, it has been "down-graded" to C where voluntary eradication applies only to particular territories that want to maintain a disease-free status.

The experience of two decades of control of BT in Europe has shown that it is not possible to stop the disease by culling infected animals or by controlling animal movements. The primary risk factor maintains the ecology of the vector *Culicoides* spp. and its wider expansion northwards for which there are presently no effective control tools.

## 4. The Way Forward

The choice of category for a disease does not necessarily relate to the costs of control but depends on the pathogen involved, e.g., the dissemination of low pathogenic avian Influenza could affect 500 million birds and will be classified in categories D–E in future Annex IV. Avian influenza is a disease subject to surveillance and movement control but will be in the same group of diseases as, e.g., ovine epididymitis. The latter only requires limited resources for interventions and will only concern the centres of breeding. Alternatively, measures could be applied to simply control the circulation of seminal material, embryos, etc. There are two cases that provide the same measures (category D–E) but involve a very different economic commitment in terms of intensity, extension, and costs of the actions. The measures will, therefore, always be economically proportionate to the impact of the pathogen in the territory: control of a category A or B disease (in an almost free country) could also be less expensive than a disease included in C, D, or E.

The provisions set out in the new Delegated Act art. 8 and 9 provide that the new intervention measures are addressed not only to species known to be the main target of a particular disease but also to all other species susceptible to infection. Thus, some diseases communicable between different domestic and wild species such as TB, brucellosis, rabies, and BT will be defined as "multi-species diseases" in this regulation and have different levels of intervention (B, C, D, and E) depending on the species involved. Bovine TB, for example, requires obligatory eradication from the European territory (cat. B) and has also been found in the genus *Capra* (specifically listed in Annex IV). TB can also infect swine and camelids to varying degrees and, thus, is subjected to category D measures. However, category E applies only if other mammals like badgers and all susceptible carnivores are at risk of infection.

These contentious issues have led to numerous meetings between the delegated experts of the various MS and experts from the Commission, OIE, and EFSA. An agreement or, at least, a compromise was finally reached between what the AHL dictate and what the 28 MS believe are achievable. This compromise, currently in operation, has often proved to be very controversial, e.g., a surveillance/control action to intervene in the beekeeping sector because of varroasis in Italy or France involves thousands of operators while, in Sweden or Finland, it will only affect a few dozen. Reasoned debates during the commissions for elaborating the new Operative Acts or debates passing from good intentions related to risk analysis to everyday reality (such as effects on markets) were tortuous, difficult, and very often hindered by national interests.

## 5. Future Situation in Italy

What will happen in Italy? Will it see the scrapping of Presidential Decree 320/1954 [6]? From the methodological process currently shared, it can be seen that the AHL leaves less room for discretion and interpretation of the law because each category provides clear and defined measures. The current Presidential Decree 320/1954 declares that all the provisions of the veterinary policies can be totally or partially implemented once an outbreak of transmissible disease is suspected. This margin of discretion has often caused doubts about how to adapt or apply sanitary actions. Unfortunately, it has actually encouraged the failure of health professionals to report disease for fear that restrictions or rules will be too severe. Thus, this causes economic hardship to the farmers, which is sometimes greater than that caused by the disease. This lack of discretion will enable operators/veterinarians to have a clearer and more defined normative reference.

In future Annex II 32, diseases have been excluded from the list of the original 62 diseases included in our regulation of Veterinary Police DPR 320 of 1954 (Chapter I—Infectious and diffusive diseases of animals subject to health measures). Those concerning small ruminants comprise:

(7) Contagious agalactia 

(16) Gastro-enterotoxyemias

(18) Pasteurellosis 

(25) Dystomatosis 

(26) Pulmonary and intestinal strongylosis 

(27) Mange 

The exclusion of these diseases from the new European regulation may be justified, on the one hand, by the lack of significant epidemiological data (for example, in Italy, only about 50 outbreaks of contagious agalactia are officially declared every year). The implications of this under-reporting are described in a recent paper on contagious agalactia as are the impacts on the exclusion of CA from the new law when it takes effect in 2012 [7]. On the other hand, some diseases with poor transmissibility and a small impact on trade will no longer benefit from European controls, such as border inspections and the requirement for appropriate travel documentation. From now, each MS will have to organize surveillance and control plans, which will almost certainly involve an excess of official bureaucracy.

These changes should, however, improve surveillance and control systems in Italy for some low risk diseases, which, up to now, have been regulated only by the scientifically obsolete Presidential Decree 320/1954. This could stimulate improved alternative and appropriate solutions for good health practices, which may provide guidelines for management of outbreaks better suited for these minor diseases. It should not be forgotten that, in the spirit of AHL, if new epidemiological evidence requires an increase in control measures for certain diseases, it will always be possible to appeal to the flexibility and revision of the annexes currently being published.

What will happen after April 21, 2021? Despite the on-going controversies, great progress has been made toward the development of a modern animal health system based on scientific knowledge and epidemiology thanks largely to the work and the mediation of various groups in the commission and an incredible effort of sharing, coordinated by the commission among the 28 MS. The Italian Department of Health, a pioneer in “One Health” responsible for both human and animal health, fully supports implementation of AHL in developing a new culture of disease prevention in Europe.

## Figures and Tables

**Table 1 animals-10-00900-t001:** Listed diseases related to small ruminants, their category, and range of target species.

Disease	Category	Target Species
Foot and mouth disease	A + D + E	*Artiodactyla*, *Proboscidea*
Infection with rinderpest virus	A + D + E	*Artiodactyla*
Infection with Rift Valley fever virus	A + D + E	*Perissodactyla*, *Antilocapridae, Bovidae*, *Camelidae*, *Cervidae*, *Giraffidae*, *Hippopotamidaae*, *Moschidae*, *Proboscidea*
Infection with *Brucella abortus*, *B. melitensis*, * B. suis*	B + D + E	*Bison* ssp., *Bos* ssp., *Bubalus* ssp., *Ovis* ssp., *Capra* ssp.
Infection with Mycobacterium tuberculosis complex (*M. bovis*, *M. caprae*, *M. tuberculosis*)	B + D + E	*Bison* spp., *Bos* spp., *Bubalus* spp.
Infection with rabies virus	B + D + E	*Carnivora, Bovidae*, *Suidae*, *Equidae*, *Cervidae*, *Camelidae*
Infection with bluetongue virus (serotypes 1-24)	C + D + E	*Antilocapridae*, *Bovidae*, *Camelidae*, *Cervidae, Giraffidae*, *Moschidae*, *Tragulidae*
Infection with epizootic hemorrhagic disease virus	D + E	*Antilocapridae*, *Bovidae*, *Camelidae*, *Cervidae*, *Giraffidae*, *Moschidae*, *Tragulidae*
Anthrax	D + E	*Perissodactyla*, *Artiodactyla*, *Proboscidea*
Surra (*Trypanosoma evansi*)	D + E	*Equidae, Artiodactyla*
Paratuberculosis	E	*Bison* ssp., *Bos* ssp., *Bubalus* ssp., *Ovis* ssp., *Capra* ssp., *Camelidae, Cervidae*
Q fever	E	*Bison* ssp., *Bos* ssp., *Bubalus* ssp., *Ovis* ssp., *Capra* ssp.
Sheep pox and goat pox	A + D + E	*Ovis* ssp., *Capra* ssp.
Infection with peste des petits ruminants virus	A + D + E	*Ovis* ssp., *Capra* ssp., *Camelidae, Cervidae*
Contagious caprine pleuropneumonia	A + D + E	*Ovis* ssp., *Capra* ssp., *Gazella* ssp.
Ovine epididymitis (*Brucella ovis*)	D + E	*Ovis* ssp., *Capra* ssp
Infection with *Burkholderia mallei* (Glanders)	A + D + E	*Equidae, Capra* ssp., *Camelidae*

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
