# Peer review of "The Implications of EU Regulation 2016/429 on Neglected Diseases of Small Ruminants including Contagious Agalactia with Particular Reference to Italy"

_animals, 2020, doi:10.3390/ani10050900_

Round 1
Reviewer 1 Report
The manuscript clearly and comprehensively explains the applications of the f the new EU regulation Animal Health Law, in particular the news related to sheep and goat diseases. The commentary will be of interest to the readers of Animals, and it is accept in the present form.
Author Response
Point one
The manuscript clearly and comprehensively explains the applications of the f the new EU regulation Animal Health Law, in particular the news related to sheep and goat diseases. The commentary will be of interest to the readers of Animals, and it is accept in the present form.
Thank you for your positive comments. We see there are no issues here
Reviewer 2 Report
In this commentary, the authors explained the European Union Regulation Animal Health Law (Reg. EU 429/2016) on transmissible animal diseases and amending and repealing certain acts in the area of animal health (“Animal Health Law”) with emphases on important diseases of sheep and goats. The implications of this law on Italy’s existing laws on animal health was also discussed.
Specific Comments
- It is not clear why the authors referred to the regulation Animal Health Law (Reg. EU 429/2016) as new. Although, it is a new law, there is only one Reg. EU 429/2016 and no older version.
- In lines 15-16, the authors referred to law being discussed as “new EU regulation Animal 15 Health Law (Reg. CE 429/2016)” but as “new Animal Health Law (AHL) EU 2016/429” in line 57. Please edit.
- Iine 123: Please remove stop sign (.) after category.
- Line 140: The experience of two decades of control of BT in Europe has shown that it is not possible to ….
- Lines 158 – 166: Kindly rephrase the sentence to make reading and understanding easier. You may consider splitting it.
- Line 178: Insert comma after shared. “….. the methodological process so far shared, it can be seen that the AHL leaves less room for discretion ….”
- Line 180: Was it meant to be “veterinary policies”?
- Lines 203-205: Insert in after “systems” and /1954 after “320”and delete in after “diseases”. “These changes should, however, improve surveillance and control systems in Italy for some low risk diseases which up to now have been regulated only by the scientifically obsolete Presidential Decree 320/1954; ….”
- Line 215: Remove full stop after (One Health)
Author Response
- Thank you for these helpful comments which improve the text significantly
- It is not clear why the authors referred to the regulation Animal Health Law (Reg. EU 429/2016) as new. Although, it is a new law, there is only one Reg. EU 429/2016 and no older version.
- In lines 15-16, the authors referred to law being discussed as “new EU regulation Animal 15 Health Law (Reg. CE 429/2016)” but as “new Animal Health Law (AHL) EU 2016/429” in line 57. Please edit.
L15 we have deleted "new"
L57 We have included "recent"
- Iine 123: Please remove stop sign (.) after category.
full stop removed
- Line 140: The experience of two decades of control of BT in Europe has shown that it is not possible to ….
we have amended this
- Lines 158 – 166: Kindly rephrase the sentence to make reading and understanding easier. You may consider splitting it.
-
Thus, some diseases communicable between domestic and wild species such as TB, brucellosis, rabies and BT will be defined as "multispecies diseases" in this regulation and have different levels of intervention (B, C, D and E) depending on the species involved. Bovine TB, for example, which requires obligatory eradication from European territory (cat. B) has also been found in the genus Capra (specifically listed in Annex IV). TB can also infect swine and camelids to varying degrees and thus subject to cat. D measures but cat. E applies only if other mammals like badgers and all susceptible carnivores are at risk of infection
- Line 178: Insert comma after shared. “….. the methodological process so far shared, it can be seen that the AHL leaves less room for discretion ….”
L187 comma inserted
- Line 180: Was it meant to be “veterinary policies”?
L189 Yes "policies" inserted
- Lines 203-205: Insert in after “systems” and /1954 after “320”and delete in after “diseases”. “These changes should, however, improve surveillance and control systems in Italy for some low risk diseases which up to now have been regulated only by the scientifically obsolete Presidential Decree 320/1954; ….”
L212-214 amendments made
- Line 215: Remove full stop after (One Health)
- L224 full stop added
Reviewer 3 Report
I am reviewed the manuscript “The implications of EU Regulation 2016/429 on 2 contagious agalactia and other neglected diseases of 3 small ruminants with particular reference to Italy” and this article falls within the scope of the journal Animals.
The article is well written, clear and the structure is adequate. These authors explain the future application of the new EU regulation relating to notifiable diseases of small ruminants and their surveillance and control.
In my opinion, the article is adequate because it´s a topic of interest considering the future management of diseases in the European Union. The characteristics of the new EU health system and the differences between the previous Italian regulations and the new law are well explained by the authors but other aspects could be looked at in depth to clarify the consequences for readers.
I propose these modifications:
-Related to the diseases covered by the new EU law, the inclusion in the new list of the exotic epizootics or other "multi-species" recognized endemic diseases as blue tongue, tuberculosis, brucellosis or anthrax is probably well understood by readers. Despite it, it could be nice to discuss the criteria used to include or not other diseases, if authors have know that information.
-About the diseases included in the new category D. Are the control movement strategies currently applied in Italy to prevent their spread?
-The future Annex II have excluded 32 diseases from the current Italian list of 62 diseases. I suggest including the list of all these diseases excluded to clarify the impact of the new law on current Italian control programs. Moreover, it could be nice to discuss this impact on current strategies conducted on Italian small ruminant herds.
-Contagious agalactia has not been included in the new EU list. Maybe because of the lack of significant epidemiological data or official declarations as the authors suggested. Despite CA outbreaks are rarely declared, but considering that CA is one of the most important diseases affecting small ruminants in Europe, with an endemic situation reported in countries as Italy, France, Spain, Greece or Portugal, this exclusion could be surprising. Do the authors consider that the exclusion of CA of the new list could be a problem in order to develop control programs in Italy or other EU countries?
Minor suggestions:
-I suggest deleting the lines 81 to 89 because the manuscript is focus on ruminant diseases.
-I also suggest deleting “contagious agalactia” in the title because the manuscript is not focused on this disease.
-Why the authors considered that Q fever or especially paratuberculosis are diseases “of less veterinary impact” ? Please clarify.
In my opinion, after the modifications, this manuscript is suitable for its publication in Animals.
Author Response
thanks for useful comments which improve the text
I am reviewed the manuscript “The implications of EU Regulation 2016/429 on 2 contagious agalactia and other neglected diseases of 3 small ruminants with particular reference to Italy” and this article falls within the scope of the journal Animals.The article is well written, clear and the structure is adequate. These authors explain the future application of the new EU regulation relating to notifiable diseases of small ruminants and their surveillance and control. In my opinion, the article is adequate because it´s a topic of interest considering the future management of diseases in the European Union. The characteristics of the new EU health system and the differences between the previous Italian regulations and the new law are well explained by the authors but other aspects could be looked at in depth to clarify the consequences for readers.
We are grateful for the positivity of the remarks
I propose these modifications:
-Related to the diseases covered by the new EU law, the inclusion in the new list of the exotic epizootics or other "multi-species" recognized endemic diseases as blue tongue, tuberculosis, brucellosis or anthrax is probably well understood by readers. Despite it, it could be nice to discuss the criteria used to include or not other diseases, if authors have know that information.
L84-85 Some further information regarding the criterion has been added
-About the diseases included in the new category D. Are the control movement strategies currently applied in Italy to prevent their spread?
Presently in Italy, like the rest of the EU, control movement strategies are focused on single diseases but this will change to disease category in April 2021 when AHL 429/2016 comes into force. We haven't modified the text as we feel this is self evident
-The future Annex II have excluded 32 diseases from the current Italian list of 62 diseases. I suggest including the list of all these diseases excluded to clarify the impact of the new law on current Italian control programs. Moreover, it could be nice to discuss this impact on current strategies conducted on Italian small ruminant herds.
There is some uncertainty in Italy as to who will be responsible for these omitted diseases when the EU law comes into force next year so we would like to leave the table out at this stage. We have included a reference (7) on the impact of current strategies on CA L208-210
-Contagious agalactia has not been included in the new EU list. Maybe because of the lack of significant epidemiological data or official declarations as the authors suggested. Despite CA outbreaks are rarely declared, but considering that CA is one of the most important diseases affecting small ruminants in Europe, with an endemic situation reported in countries as Italy, France, Spain, Greece or Portugal, this exclusion could be surprising. Do the authors consider that the exclusion of CA of the new list could be a problem in order to develop control programs in Italy or other EU countries?
Reference 7 includes reference to discussion on the impact of exclusion (L208-210)
Minor suggestions:
-I suggest deleting the lines 81 to 89 because the manuscript is focus on ruminant diseases
We think this provides important background information so would like to keep this section.
-I also suggest deleting “contagious agalactia” in the title because the manuscript is not focused on this disease.
We have modified the title to "The implications of EU Regulation 2016/429 on neglected diseases of small ruminants including contagious agalactia with particular reference to Italy" to provide an important example of the neglected diseases
-Why the authors considered that Q fever or especially paratuberculosis are diseases “of less veterinary impact” ? Please clarify.
Epidemiological data from EU shows these diseases have a low impact within Europe and are generally easily treatable especially for Q fever L97-98
In my opinion, after the modifications, this manuscript is suitable for its publication in Animals.